# Ambulance use for 'primary care' problems: an ethnographic study of seeking and providing help in a UK ambulance service

Matthew James Booker, Sarah Purdy, Rebecca Barnes, Ali R G Shaw

Centre for Academic Primary Care, Population Health Sciences, University of Bristol, Bristol, UK

**Correspondence to**
Dr Matthew James Booker;
Matthew.Booker@Bristol.ac.uk

## ABSTRACT

**Objectives** To explore what factors shape a service user's decision to call an emergency ambulance for a 'primary care sensitive' condition (PCSC), including contextual factors. Additionally, to understand the function and purpose of ambulance care from the perspective of service users, and the role health professionals may play in influencing demand for ambulances in PCSCs.

**Design** An ethnographic study set in one UK ambulance service. Patient cases were recruited upon receipt of ambulance treatment for a situation potentially manageable in primary care, as determined by a primary care clinician accompanying emergency medical services (EMS) crews. Methods used included: structured observations of treatment episodes; in-depth interviews with patients, relatives and carers and their GPs; purposeful conversations with ambulance clinicians; analysis of routine healthcare records; analysis of the original EMS 'emergency' telephone call recording.

**Results** We analysed 170 qualitative data items across 50 cases. Three cross-cutting concepts emerged as central to EMS use for a PCSC: (1) There exists a typology of nine 'triggers', which we categorise as either 'internal' or 'external', depending on how much control the caller feels they have of the situation; (2) Calling an ambulance on behalf of someone else creates a specific anxiety about urgency; (3) Healthcare professionals experience conflict around fuelling demand for ambulances.

**Conclusions** Previous work suggests a range of sociodemographic factors that may be associated with choosing ambulance care in preference to alternatives. Building on established sociological models, this work helps understand how candidacy is displayed during the negotiation of eligibility for ambulance care. Seeking urgent assistance on behalf of another often requires specific support and different strategies. Use of EMS for such problems—although inefficient—is often conceptualised as 'rational' by service users. Public health strategies that seek to advise the public about appropriate use of EMS need to consider how individuals conceptualise an 'emergency' situation.

## INTRODUCTION

Emergency medical services (EMS) calls have been rising in the UK over recent years at 7% per annum.[1 2] Increasingly, these calls are for

### Strengths and limitations of this study

► This is the first time that such a range of complementary data sources have been used to explore 'primary care sensitive' conditions (PCSCs) in the ambulance service in such case-level detail, offering new insights from multiple perspectives on the same encounter.
► The study draws on a relatively small number of cases in a single service, and the methods of eligible case identification necessarily have some subjectivity.
► Despite this, regular study advisory group scrutiny and a considered, reflexive approach in the analysis provides confidence that the cases and phenomena described are 'typical' and yield more nuanced new insights on the classical medical sociological models of 'help-seeking'.

conditions or situations that could potentially be managed through a timely contact with a primary care provider.[2] Indeed, recent UK evaluations suggest only approximately 10% of calls represent immediate life-threatening medical emergencies.[3] So-termed 'primary care sensitive' conditions (PCSCs)—which include some social situations and mental health problems—often represent less efficient use of ambulance resources, and may result in patients requiring a multitude of contacts to resolve their need.[4]

Despite UK policy favouring an integrated urgent care service that more closely matches 'response' with 'request',[5] relatively little depth-work has considered how and why PCSCs reach ambulance service workflows. A recent systematic review[6] and evidence synthesis[7] identified that the emotional impact of needing advice 'urgently' may shape the choices made when help-seeking, offering a more nuanced understanding of the classic illness models.[8] This work has also highlighted the role that certain sociodemographic factors play, some of which appear

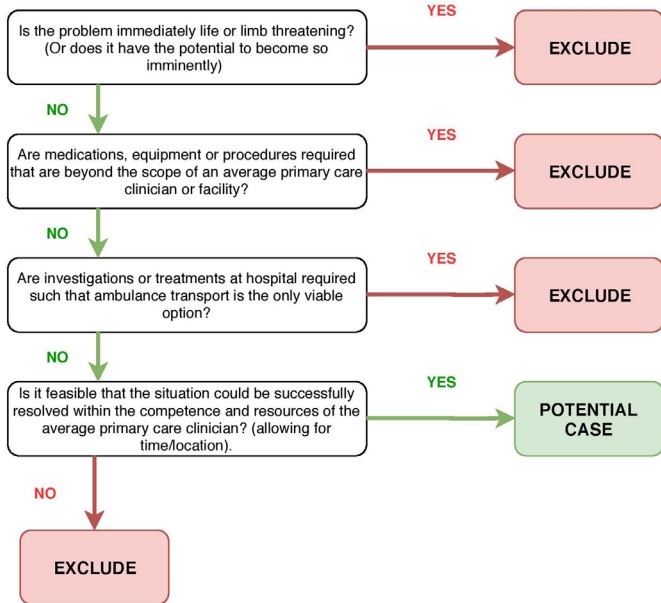

**Figure 1** Indicator criteria for 'primary care sensitive' case identification.

internationally universal in the context of avoidable ambulance use.[6] Previous interview studies (eg, Booker *et al*[9]) have offered some insights into service users' experiences of ambulance care for PCSCs. This includes difficulties accessing services and confusion about how services are structured—findings which have been mirrored more generally in the urgent care, GP out-of-hours and emergency department settings.[10–12] However, there remains a fairly superficial understanding of how all contributing factors—personal, situational, professional and institutional—combine to reflect the observed trend in increased ambulance attendance for PCSCs.

Ethnography has recently been applied to the study of interactions between ambulance clinicians and patients.[13] By employing the principles of 'triangulation',[14] it is possible to use a variety of qualitative data, collected from complementary perspectives, to offer a much richer understanding of a phenomenon. This ethnographic study, therefore, sought to employ multiple methods to explore how and why an exemplar set of PCSCs ended up receiving ambulance treatment. Ultimately, the study aims to improve understanding of how to meet these needs.

## METHODS
### Participants and setting
This study took place in one UK Ambulance Service during a period of 5 months, spanning September 2016 to January 2017. The UK is divided into 13 regional ambulance services (with additional, separate provision for those islands with autonomous administration). The service participating in this study handles approximately 250 000 emergency calls per annum, and serves a population of just under 3 million people across a geographic

area exceeding 20 000 square kilometres. Cases were eligible for inclusion in this study if the following criteria were met:

► The patient was an adult with capacity to consent to study participation;
► The caller (either the patient or their representative) had dialled the national emergency '999' number and asked for an ambulance;
► The call had been triaged to receive an emergency ambulance response (of any priority);
► The reason for their call was subsequently deemed to be for a potentially 'primary care sensitive' situation.

Such 'primary care sensitive' situations were identified by the first author—MJB, a primary care clinician researcher—who accompanied front-line ambulance crews during routine shifts in a 'non-participant observer' capacity. A set of consensus-informed indicator criteria (figure 1) and professional judgement were used to identify potential cases. Conditions and situations that would likely be realistically amenable to resolution in a primary care setting were considered eligible. This method of identifying 'primary care' cases was favoured over attempts to use clinical records or routine outcome data, as it was felt that a primary care clinician working at the scene could more accurately assimilate all of the clinical, situational and contextual nuances in real-time to make a judgement. The basis for each recruitment was discussed and agreed at regular study team meetings during the recruitment phase, with recruitment continuing until a broad and diverse representation typical of 'urgent primary care' presentations had been included, as determined by consensus with the study advisory panel. This panel comprised social scientists, a GP, a paramedic and a patient/carer representative.

At the conclusion of the ambulance service treatment, the patients (and/or their proxy callers, where appropriate) were provided with information regarding the study. Patients or carers who made contact to request further details were subsequently formally consented. In cases where someone other than the patient had made the 999 call, consent was sought from the caller as well as the patient.

### Patient and public involvement
The study team regularly consulted with an Urgent Care Service Users study advisory panel, including patient and carer representatives who had recently accessed ambulance care. This group helped shape the design and focus of the study, ratify the research questions, advise on the content of participant-facing study literature and refine the dissemination strategy.

### Data collection methods and sources
For each treatment contact observed, MJB completed an ethnographic template according to the nine observational dimensions of Spradley,[15] (which are now established as key domains for ethnographic studies of healthcare encounters[16]). This template included details

on (among others) the space, setting, participants, activities, objects and emotions evident in the encounter. Detailed field notes and a reflective diary supplemented these. These were complemented by ethnographic interviews[17] with patients and—where possible—any relatives or carers present, which were conducted within 14 days of treatment, securely audio-recorded, transcribed verbatim and checked for accuracy by participants. Where the observation or interview indicated there may be value in further insights from in-hours primary care, GPs were approached by letter to participate in a semistructured interview,[18] recorded and transcribed as above. These interviews were supported by a printout of the last 12 months of primary healthcare records as stimulus material, used to inform prompts during interviews. Ambulance clinicians consented to be observed at the start of the shift, and to the making of secure audio recordings of spontaneous 'professional conversations'[19] throughout the shift. These were subsequently transcribed verbatim and matched to the cases. The original 999-call recording was securely obtained from the ambulance service, redacted and transcribed according to the conventions of Conversation Analysis (CA).[20] A more detailed CA-based analysis has been performed on these recordings and is reported elsewhere.[21] For the purposes of this study, a 'realist' content analysis approach was used to enable comparisons to be made across other data sources.

## Rationale for an ethnographic approach

Within the field of applied health research, ethnography has come to encompass a range of complementary, overlapping qualitative principles and techniques that may include the concepts of 'case studies' or 'life histories', constructed through fieldwork undertaken over time among the people of interest.[17] Ethnography involves the telling of 'credible, rigorous and authentic stories from the perspectives of people experiencing the phenomena of interest in the context of their daily lives and culture'.[22] Features of an ethnographic approach include: a strong emphasis on exploring the nature of a social problem; a tendency to work with unstructured data; investigation of a small number of cases in great detail; and analysis that seeks to interpret the meaning and functions of human actions within a specific context.[23] The key principles of the ethnographic approach, drawing upon the epistemology of subtle (critical) realism,[24] are therefore well suited to exploring the mixed physical, social and psychological manifestations of 'unwellness' in the prehospital setting, and understanding the actions people take to secure urgent advice.

## Ethical considerations

Due to the nature of the possible 'urgency' of the treatment contact, it was not practical to obtain full informed consent for the ethnographic observation at the outset, and in practice, some data were necessarily collected (in the form of field notes and observations) before consent was achieved. At the earliest practical opportunity, the observing primary care clinician researcher was introduced to the patient and verbal consent sought to remain. A 'shared understanding' document served as an advance agreement between the researcher and the ambulance crews, such that if any circumstances arose where it was felt that it was either unsafe or inappropriate for the researcher to remain, a process was in place for withdrawal and deletion of any data. The study received a favourable opinion from South West (Frenchay) Research Ethics Committee (Ref 15/SW/0307).

## Data analysis

Analysis commenced early during data collection and continued throughout, following an iterative-inductive approach. The overall analysis approach was thematic, informed by the principles of constant comparison.[25] An individual patient with all their associated data was treated as a 'case'. First, within-case analysis was conducted to capitalise on the rich casewise ethnographies. Second, across-case analysis sought to develop an understanding of common phenomena across the whole dataset.

Data pertaining to each specific case were indexed and collated with the assistance of the qualitative analysis software NVivo (V.10). Interview transcripts, field notes, conversations, ethnographic frameworks and 999-call transcripts relating to each case were treated as separate data items. Each data item was repeatedly read and re-read to build familiarity, and then first-level coded, using 'free-form' open codes. Primary care records were similarly first-level coded, in a manner informed by Document Analysis (a specific form of Content Analysis that treats the record as a 'document with a specific purpose'[26]).

The codes from these separate data sources were then combined to develop a set of second-level axial codes pertaining to all pooled data items about an *individual* case. A third tier of coding combined these axial codes into themes. In this analysis, the term *theme* is used to refer to patterns that run within a case. The techniques of charting aided this process.[27] figure 2 provides an illustrative example of within-case charting of themes.

Second, to identify and explore issues across and between cases, a final level of coding sought to combine these *themes* into *cross-cutting concepts*. It is recognised that the term 'concept' has a variety of uses and meanings in the social sciences. In this analysis, the term *concept* is used to refer to a high-level phenomenon that runs among and between cases. figure 3 provides a diagrammatic overview of the relationship between data, cases, themes and concepts.

## RESULTS

A total of 180 hours of observation were completed, as summarised in table 1. This generated 170 data items across 50 cases (48 ethnographic observation templates, 44 patient interviews, 18 carer interviews, 8 GP interviews, 8 ambulance staff conversations, 10 primary care record

| | Case A | | |
|---|---|---|---|
| **Data Source** | **Axial Code:** *Discounting of alternative sources of help* | **Axial Code:** *My burden of health problems makes access difficult* | **Axial Code:** *No one understands what it is like to live with my health problems* |
| **Observations** | First-level code: Repeatedly expressed that GP wouldn't be able to help with this problem [A.16B] | | First-level code: List of medications, diagnoses and specific problems these cause carried in handbag [A.112J] |
| **Patient Interview** | First-level code: Difficult to make oneself understood over the telephone cf face-to-face [A.23G] | First-level code: Breathlessness makes getting myself to the treatment centre impossible [A.72R] | First-level code: Difficulty summarising how the condition makes me feel to health professionals [A26.Y] |
| **Carer Interview** | First-level code: Speaking to the doctor hasn't been able to resolve this previously [A.63F] | | First-level code: Need to explain on behalf of patient as finds upsetting to talk about [A.83.Y] |
| **Primary Care Records** | | First-level code: Records annotated to allow telephone requests for repeat medication. [A.11K] | |
| **GP Interview** | | | First-level code: Depression largely results from severity of illness [A.4.J] |
| **Ambulance Clinician Conversation** | First-level code: Patients give reasons why they have not accessed care down another avenue to justify call [A.12H] | | First-level code: Patients struggle to explain what prompted the call *today specifically* in an on-going longer term problem [A.67.B] |
| **Field Note Diary** | First-level code: Justification for 999 call made on basis of exclusion of other viable options [A.55A] | | First-level code: Difficultly communicating how challenging day-to-day life is [A.53.K] |

**Figure 2** The 'charting up' process used to analyse data sources within cases.

extracts and 46 999-call recordings). The characteristics of cases are shown in table 2.

Three cross-cutting concepts emerged from the cross-case analysis. These are:

1. There exists a typology of circumstances that result in an ambulance for a 'primary care' problem. These circumstances result from both internal patient-specific factors and external environmental factors.
2. Calling an ambulance on behalf of someone else generates a specific anxiety around prioritisation and urgency.

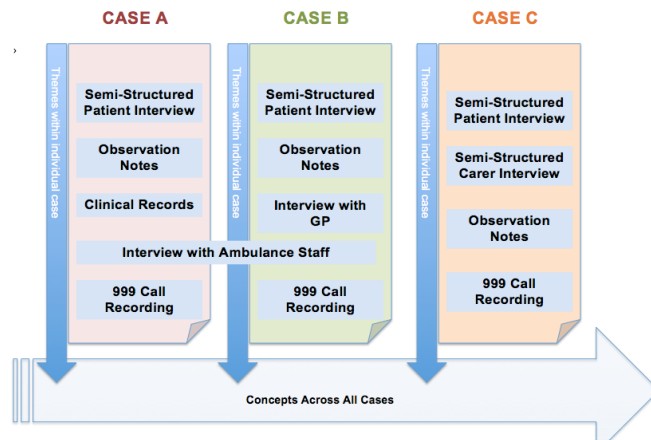

**Figure 3** The relationship between cases, *themes* and *concepts*.

3. Clinicians are conflicted about dealing with the problem in front of them, and fuelling further demand.

## There exists a typology of circumstances that result in an ambulance for a 'primary care' problem

This concept groups together and describes the circumstances that appear to result, most fundamentally, in the trigger to make contact with the ambulance service.

These sets of circumstances can be considered together as a 'typology' of triggers. Although it can not necessarily be claimed that this group of trigger circumstances is true for *all* '999' calls to the ambulance service, within this group of cases, it is possible to summarise all of the circumstances under nine headings. They have been classified as either 'internal factors' or 'external factors' (or both). 'Internal factors' tend to describe a participant's perception of their lived experiences. 'External factors' describe the actions and perceptions of people or services around the patient. This classification helps to typify some of the circumstances and there is overlap—contradicting examples are highlighted below, where they occur.

Importantly, this typology of 'trigger factors' appears consistent in shaping both a patient's decision to call an ambulance, and a carer or relative's. This would suggest that these factors do more broadly describe the *circumstances* rather than the *individuals* involved. Table 3 summarises these trigger factors.

### An arbitrary deadline is reached (internal factor)

This classification was a common trigger for patients with both acute conditions and long-term problems, and describes a circumstance whereby the patient or carer sets an arbitrary time frame for resolution of some symptom or situation. If that time frame is exceeded, the patient reaches the conclusion that the situation justifies an ambulance call:

> I'd been on these things [antibiotics] for two days by that time, and I hadn't seen signs of improvement. He'd told me that if the redness spreads across the line to call him back. Well it hadn't done that. But he also said it would start to get better in a couple of days. I took the first one with tea on Tuesday, so, well, it was two full days by teatime Thursday wasn't it?
>
> Patient interview, case 31 (cellulitis).

The ambulance staff appeared very familiar with this situation, and reflected how it even influences the organisation's operational planning:

> Yeah, people do that don't they? They sort of set a line in the sand around key points of the day? We find that a lot. For some its dinnertime or bedtime or whatever. The service does see increases in calls around certain specific times of the day because of people doing that. I suppose it is only natural that you draw a line in the sand at a specific point but I do struggle to understand the decisions sometimes.
>
> Ambulance staff, conversation 1

**Table 1** Spread of observation hours according to crew type, time and day

| Characteristic | Shift hours (in rural setting) | Shift hours (in urban setting) |
| --- | --- | --- |
| Solo paramedic responder (rapid response vehicle) | 24 | 24 |
| Dual-crewed paramedic ambulance | 56 | 76 |
| Daytime (08:00-20:00) | 44 | 76 |
| Night time (20:00 – 08:00) | 36 | 24 |
| Weekday | 60 | 76 |
| Weekend | 20 | 24 |

For these patients, the timeframe of their experienced illness appears the principal driver in making the call.

### The situation becomes 'overwhelming' (internal factor)

This classification describes a situation whereby the caller feels that all their current issues—symptoms, social circumstances, emotional resilience—have reached a point that they cannot continue to function without ambulance help. The term 'overwhelming' was drawn from the following participant interview:

> I was just completely losing track of it all to be honest. The pills I had to give [my wife], all the comings and goings of the carers, the dressing kept coming off, the phone is going all the time, I need to do her tea and sort everything at home out, and then this? This mix-up with the medicines. Truth be told I just felt a bit overwhelmed by it all, you see?

> Carer interview, case 36, Medication administration error.

There are examples in the case set of where patients themselves feel overwhelmed and where carers or relatives feel overwhelmed. Although there is a link with the concept of isolation, being 'overwhelmed' does appear conceptually distinct, as some non-isolated callers also felt 'overwhelmed' by the burden that their experience placed upon them.

### A symptom triggers a 'red flag' (internal factor)

In a number of cases, patients or carers had been managing their illness or condition up to a point where a new feature or symptoms emerged that triggered concern about a serious illness. Healthcare professionals often refer to 'red flag' symptoms as those that may be indicative of a serious underlying illness, and that warrant being taken seriously. It appears this term—both literally and conceptually—has entered the patient lexicon, too, as it was referred to as the specific trigger in a number of cases:

**Table 2** Characteristics of recruited 'cases'

| Characteristic | Cases (n=50) |
| --- | --- |
| Mean age (years) | 57.4 |
| Age range (years) | 18 – 92 |
| Female | 30 (60%) |
| Has a formal carer | 18 (36%) |
| Not the patient making the 999 call | 31 (62%) |
| Clinical problem | |
| Acute infection | 7 |
| Breathing problems | 5 |
| Mental health problems | 5 |
| Abdominal Pain | 4 |
| Falls, faints & funny turns | 4 |
| Sickness/gastroenteritis | 3 |
| Confusion | 3 |
| Other | 3 |
| Chronic pain condition flare-up | 3 |
| Urinary symptoms | 2 |
| End of life / palliative care problem | 2 |
| Chest pain | 2 |
| Musculoskeletal pain | 2 |
| Skin problems | 2 |
| Headaches | 2 |
| Medication problems | 1 |
| Outcome | |
| Transported to hospital | 14 (28%) |
| Treated at scene—no referrals | 13 (26%) |
| Treated at scene—referred to GP | 18 (36%) |
| Treated at scene—referred to community nursing or social care | 4 (8%) |
| Refused further treatment | 1 (2%) |

**Table 3** Trigger factors that result in an ambulance contact

| 'Internal' factors | 'External' factors |
| --- | --- |
| An arbitrary deadline is reached | An outsider offers advice/an opinion |
| The situation becomes 'overwhelming' | An alternative avenue of care meets a block |
| A symptom triggers a 'red flag' | A healthcare professional takes charge |
| Experience of isolation | The problem belongs to someone else |
| A change occurs in care provision | |

Well, when he had chest pain too, you don't ignore that do you? It's like some kind of red flag to a bull isn't it? You act - you call - huh?

Carer interview, case 8, muscular chest pain

I thought 'oh my God you get a rash in meningitis' don't you? Don't you?

Patient interview, case 19, skin complaint

Such discussion of 'red flags' was evident in some of the primary care consultation records, as part of the safety-netting process:

SOS and red flags disc[ussed]. Knows [to call] OOH [out of hours]/999 etc if ^pain/haemop[tysis]/SOB[shortness of breath] etc at any further stage.

Primary care record extract, case 35, swollen leg.

The vagueness of the clinician's advice about time-frames within which patients should act if they experience a 'red flag' was cited as a particular source of anxiety in some patient and carer interviews, and was reflected in the observations. There were also examples of patients attempting to self-manage conditions through internet research, and misattributing a description of a 'red flag' to their own situation.

### Patient experiences isolation (internal factor)

The experience of isolation appeared to drive contact with the ambulance service in a number of ways. There were examples of cases where the isolation was to do with very practical aspects of living, possibly sudden or abrupt—perhaps someone who usually provided counsel or a channel of connection was no longer available. There were other examples where the isolation was long-standing, with a strong social and emotional component:

[The] Lady asked me to pass cheque-book sized photograph album that was sitting on the mantelpiece next to her armchair, so she could put [it] in her purse to take with her [to hospital]. Asked her about it; noted it was embossed with the title 'friends who have entered the everlasting'. She told me how she took the memories of her friends with her wherever she went so that they were "always by her side". Asked her if there was anyone we should let know she was going in. 'No, there's no one left'.

Extract from ethnographic framework and field note diary, Case 42, Chest Infection

Interestingly, there were also examples of where isolation was expressly recognised as a feature in the lives of some participants, but rejected as a factor in triggering an ambulance contact. In these cases, participants explained how isolation was something that they had learnt to adapt to such that it was not the driving factor in seeking ambulance help:

I know I am all on my own here, and my family are far away. They can't do much practically for me. But I have found ways around that, you know? I save things

up to tell them. I know they care from afar and so I just get on with doing what is necessary rather than relying too much on them… practically, at any rate. I get my own help if I need it. It doesn't bother me that they are not on the doorstep.

Patient interview, case 21, urine infection

This particular case is interesting, as much of the interview focused on how well he felt he was coping on his own without practical support. While this participant would certainly not define himself as emotionally isolated, many of the codes in the data about this case pertained indirectly to issues of practical isolation, and so the concept of isolation was very strongly expressed in the analysis, even though he overtly rejected it.

### A change occurs in care circumstances (internal and external factor)

This category describes situations where a (usually sudden) change in the social care provision propels the patient towards ambulance care, or results in the carer calling an ambulance. The former appears most commonly due to the turmoil that the destabilising effect of carer change has:

And it was a new woman? And I don't think she got it, she didn't really seem to see how unsettled he was in and that wasn't normal for him. So I didn't think she really knew what to do. She didn't know him before, that was the trouble, so I had to act!

Carer interview, case 34, (confusion)

In this example, the change in carer provision had caused an upset to the usual routine. Deeper analysis of the 'change of carer' concept reveals that this is actually quite complex. There is a lack of familiarity, with all of the personal relationship and trust issues that this may bring. There is also a lack of familiarity—as exemplified here—with the patient's usual 'baseline' level of functioning. This can either create a situation of heightened anxiety (in the cases where someone actually appears quite unwell, but this is their normal level), or a perceived lack of awareness of subtle but important signs of deterioration. In the above example, neither the practical nor emotional benefits of familiarity were present, and the situation reached a flash point.

A change in the informal care arrangements, such as occurred when a relative became unavailable, also had a destabilising effect:

Care plan noted: 'Mr Xs son away at the moment, seems to be causing some distress and concern. Phoned son and message left to say to call dad ASAP.' The carer seemed to feel that Mr X was very unsettled by the fact his son was away.

Ethnographic framework and field note diary, case 32, unsteady on legs

### An outsider offers advice (external)

This was a common trigger, and was found to a greater or lesser extent in nearly half of all the cases. Typically, the

'outsider' was a friend or relative who offered a perspective that *increased* the perceived urgency or legitimacy of the situation. In several situations, it appeared that the 'outsider' was actually the main driving force behind the contact with the ambulance service:

> The friend who is present appears very keen for us to see the photographs of a meningococcal rash that she has brought up on her iPad, showing everyone present several times during the treatment contact that is what she thought the rash was.
>
> Ethnographic framework and fieldwork diary, case 19, rash

Ambulance staff spoke about their awareness (and even frustration) about the role that those around the patient can have in driving the situation:

> You sometimes have to just remove people… physically remove people… from the situation because they are not being helpful. It is like they are projecting their own anxieties on to the patient and it's not helping. You are trying to have a sensible chat with the patient about your clinical rationale and diagnostic reasoning, and they keep chipping in something really unhelpful… like that lad earlier who kept talking about brain tumours, yeah? I mean, that's something clearly he has got some specific issues about from his past, but it's not terribly helpful and it totally clouds the person's thinking when that is going off in their ear, you know?
>
> Ambulance clinician conversation 8, case 47, headache

The ambulance staff recognise that the driver behind certain callouts has not come from the patient themselves, and so they sometimes find themselves managing two problems—the actual clinical problem in the patient, and a separate situation in the 'other' person who is really the root of the call to the ambulance service. Staff, therefore, can feel a mixed responsibility as to whom it is they are really there to help.

### An alternative avenue of care meets a block (external factor)

This set of circumstances describes the perception of a 'road block' when trying to access care via alternative avenues. The result is that the caller feels (accurately or otherwise) they have exhausted all other options, and the ambulance is the only viable pathway. Sometimes the block is overt:

> I mean I tried that [calling the GP surgery] but they told me I should phone an ambulance

Sometimes the 'block' is less clear. In the following example, the response from mental health services is perceived as a 'block' because it did not mean the timescale that the caller has determined is appropriate for the pressing and immediate needs:

> They sent me away, said he can't see a psychiatrist until Thursday. Well that's no use is it, fobbing me off with an appointment in two days time? What do I do for the next two days, lock him in the house?.
>
> Carer interview, case 10, mental health crisis

Healthcare staff appeared aware of how this 'block' can be perceived, and that it can have consequences with regards to how patients choose to access care:

> Hmm, we try and facilitate a GP call-back, but it is often not immediate—there is often a delay once the request is passed from the reception girls. I guess that delay… we try and minimise the delay for the patients we know… but I guess that delay for some people is too long to be hanging in the air not knowing what to do? Often you phone back… like here… half an hour later and they say 'oh we've called the ambulance, don't worry now'. It is a bit frustrating.
>
> GP interview

### A healthcare professional takes charge (external factor)

This classification describes circumstances whereby a health professional takes over and directs the patient to call an ambulance. This appeared to happen in one of two ways. This might be through specific follow-up advice to the patient to take that course of action in the here-and-now as documented below:

> COPD, still exacerbating, started rescue pack, sounds SOB, advised 999
>
> Primary care record extract, case 43, COPD exacerbation

Or by offering safety-netting advice to help them identify the need to seek further medical help:

> Adv[ised to call] 999 if any change at all, if any further prob[lem]s or deterioration
>
> Case 22, Clinical records, COPD exacerbation

If the advice was delivered to the patient as recorded here, it would seem very clear that the health professional was guiding the patient towards accessing ambulance care in virtually any non-specific circumstance other than noteworthy and rapid clinical improvement. The potentially all-encompassing phrase 'if any further problems' has been found to be commonly used by GPs as a form of diagnostic safety-netting.[28]

### This problem belongs to someone else (external factor)

The final classification related to situations where the caller felt that the problem or circumstance they found themselves in was someone else's responsibility to manage. This occurred frequently where formal care staff were concerned:

> So, I am not a medical person. I cannot be making decisions about when clients should and shouldn't see doctor, mmh? If they fall, and even if no obvious injuries or pains, they need to be checked, because

I will be told… it is not my job to know if they are hurt or something. They need medical people for this. If something should happen, well - hah! I will be blamed.

Carer Interview, case 13, dizziness

Issues of accountability, (limits of) profession roles and the potential for blame and repercussions run through this example. There were also examples of a genuine desire to ensure that something wasn't missed:

Somebody needs to come and manage this. There is a process I am sure, so my role is to let them know and get that process going.

Carer interview, case 5, end of life

The field notes were also able to add an interesting perspective on this 'handing over' of the problem, but showing how some people assumed a certain course of events (eg, admission) would occur almost without question once they call the ambulance:

By the front door were two nearly packed overnight bags, a mobile phone with charger, several books, toiletry bags, and a completed 'tick sheet' of jobs including 'cancel milk, call neighbour re cat, thermostat down'. It almost appeared like a list one would write before going on holiday. It was apparent that the patient was very much expecting to be taken to hospital.

Field notes, case 6, abdominal pain.

### Calling an ambulance on behalf of someone else generates a specific anxiety around prioritisation and urgency

This concept is about seeking ambulance care on behalf of someone else, and contrasts specifically with the process of seeking ambulance care for oneself.

Within this cross-cutting concept are themes relating to responsibility. Responsibility appears to be interpreted differently, depending on whether the carer is a formal 'professional' carer, or an informal relative carer. The two groups appear to handle their perceived responsibilities differently. While there are links with the above ideas of triggers—particularly the 'the problem belongs to someone else' external trigger—this concept explores the deeper reasoning people undergo to reach that conclusion.

Formal, professional carers appear to handle their responsibility in terms of a professional duty and accountability. They see 'risks' in the terms of the potential professional consequences for them if they are viewed as having failed to do their job properly. This may lead to a lower threshold to call an ambulance:

We have to escalate, because, we could get in trouble if it is something serious and we didn't act. You have your registration to think about, and the [professional] code [of practice]. The code says you must escalate your concerns quickly.

Carer interview, case 49, confusion.

Ambulance staff also described how they notice a specific decision-making process in professional carers:

You know they wouldn't call you if it was their relative in that situation! They have their box to tick… their checklist I guess. They would clearly manage the same problem very differently if it was their mum, but it's not their mum? It's their client, or their customer, or whatever term they use. It's a different relationship and it means they act differently. They take the path of least risk I think, and that's calling us.

Ambulance clinician conversation 6

This is in contrast to the much more emotional response that many informal carers had towards decision-making and risk, seeing their responsibility much more along the lines of doing the best they could for the person they cared for:

I blamed myself for the whole mix-up really. It was up to me to put it right, do right by him, you see? I felt I had in some way caused… well not caused it but, you know, made the situation a bit more muddled, and so the right thing to do by him was to get some advice as quickly as possible. I'd owe him that at least!

Carer interview, case 40, end-of-life/medication confusion.

It appeared that for the relative-carer group, the immediacy and the urgency of the response fulfilled a very important role. They appeared to be discharging their sense of responsibility through the perceived speed of the response (and therefore how seriously they felt their request for help was being taken). When evaluating oneself, one has the advantage of experiential knowledge of 'knowing how you feel'; in contrast, one is constrained by the quality and extent of communication from another when evaluating the health state of others, which may contribute to a lower risk threshold:

She was just shaking, I didn't know what was going on! Shaking like that! [gestures]. She couldn't really tell me how she was feeling. When it's you, you know how you feeling don't you? You know if you feel unwell with it? Or if you think it's something serious? But with her…well she couldn't tell me, and so I just thought… well I didn't know, so I called the ambulance.

Carer interview, case 2, urinary tract infection

### Clinicians are conflicted about dealing with the problem in front of them, and fuelling demand

As all of the cases included in this analysis were for problems that could be deemed 'primary care sensitive', there was an inherent element of balancing the need to manage the situation that resulted in the call, and re-direct the patient to another provider:

They [the patients] want a consultation, they want to discuss their options, the pros and cons of each and

be assisted towards a decision. Well, that's 'primary care', that's not what ambulances traditionally do. They want something from the service that it is not designed or able to deliver.

GP interview

It's really hard – you know that this person has totally called the wrong people. You offer strong words of advice, but how far do you go? The person still needs treatment, so if you don't deal with the situation you just pass the person around and around.

Ambulance staff conversation, mental health condition

There is evidence in the data to suggest that patients can sometimes sense themselves that they are 'caught up' in this dilemma, and that their requests and needs can present health professionals with difficulties. One patient who felt conflicted about the best course of action showed a particularly insightful example of this. She recognised that her own limitations in determining how urgent her case was could pose a problem for the clinicians at her surgery, as she was not able to articulate with confidence a response to some of the triage questions:

The question… it's the questions they ask that are really hard to answer, you know? They say 'oh is it an emergency?' and I sometimes feel like saying 'I don't know, that's why I want to talk to the doctor! You know? I don't know. It seems silly, I mean – I know they have to ask but when you say you don't know… it sometimes feels like you are not being terribly helpful, but you don't know! And so you wonder if it is better not to get them into that pickle by just going for the ambulance you know? And then you have not had to make the situation for them [the surgery]. It is as if things are set up to take you down a certain path, you know?

Patient interview, case 22, COPD

## DISCUSSION

This study sought to further understand why PCSCs result in contact with ambulance services, by characterising the context and purpose of the request for help from the service user's perspective, and identifying if (and how) the response to that request meets that need. In order to request ambulance treatment, callers must view themselves (or the person in their charge) as 'candidates' for such assistance.

This notion of 'candidacy' describes how service users embark on negotiations with healthcare professionals (or institutions representing healthcare, such as EMS), based on their perceived entitlement to urgent care.[10] With regards to PCSCs, this study suggests that entitlement is realised through (1) experienced health state, (2) a personal assessment of risk and (3) external triggers. Importantly, this study suggests that the 'trigger

factors' outlined in cross-cutting concept one may de facto engender a sense of candidacy for ambulance care.

Outside of the context of needing urgent advice, patients and their carers are able to rationalise what 'reasonable' use of resources looks like.[29] Yet, in the heat of the moment, the influences of uncertainty, a sense of responsibility for the welfare of another and a knowledge that the system needs certain information to prioritise requests made of it, a new rationality exists. The logic behind such rationality is often presented by callers in terms of why other avenues have met 'a block'—a perception that may not always be accurate.

The seminal sociological illness-behaviour and help-seeking models have long recognised how broadly discrete 'triggers' can drive a decision towards consulting behaviour, through temporising of symptomatology (eg,[30]), the interference of symptoms with personal or vocational activities (eg,[31]), or the occurrence of an interpersonal crisis as a result of illness (eg,[31]). The influence that others have on this decision-making has also been well established, including how the so-termed 'lay referral system' is often a patient's trusted source of advice on if, when and how to consult (eg,[32]). The sanctioning of consulting decisions by trusted others is also a well-recognised aspect of primary care help-seeking behaviour.[31] This study supports the applicability of these principles to seeking ambulance care for PCSCs. Although these others may be 'outsiders', they are often seen as 'insiders' by callers, and as such their advice may be seen as more relevant than the generic institutional messages intended to mitigate demand.

Additionally, the sense of distancing oneself from one's actions is achieved through the justification of circumstances as an *emergency* situation, which is often indistinctly blended with an *uncertain* situation. This justification is—at least in part—compounded by the healthcare provider's conflicted stance on dealing with the problem *now*, or re-directing the patient to a primary care provider. There therefore exists a circular challenge—by not resolving the issue during the EMS contact when it would be technically possible to do so, the problem is perpetuated within the system. This lack of resolution is professionally unfulfilling and inefficient, yet resolving the contact feels to practitioners like reinforcement of (questionable) candidacy.

As such, practitioners offer (and service users value) other elements rather than just medical treatment. This study supports previous work, suggesting that these elements include reassurance,[33] empathy,[34] and a sense of bringing control to an unmanageable or intolerable situation.[7 9 33] The findings of this study suggest that service users might be seeking these non-medical elements of care when they make contact with the EMS for PCSCs. Indeed, irrespective of the true clinical severity of the situation, this study supports the idea that people feel at the limit of their ability to cope with the situation as they perceive it when they call—they have arrived at their own 'critical situation'.[35] The present triage processes they

encounter are neither designed—nor always able—to offer resolutions.

For nearly two decades, the academic discourse has sought to challenge the labelling of callers such as those in this study as merely 'inappropriate' users of ambulance services.[36] Indeed, international researchers are now recognising that these 'inappropriate' contacts provide useful insights into equality of access and utilisation of preventative healthcare in the community.[37] Nevertheless, the influence of healthcare professionals' views on what is 'appropriate' ambulance work continues to influence how practitioners manage these contacts.[38] Consequently, the debate about what is fuelling society's apparent general declining ability to tolerate 'uncertainty' and 'risk' continues. The established sociological concept of an increasingly 'risk averse' society[39 40] is omni-relevant. Additionally, it is important to understand that healthcare institutions display their own attitudes to 'risk' via the triage processes they require callers to undergo. This will impact on a process that is already emotionally charged.[41] Where third-party callers are involved, the projection of candidacy discussed above may be particularly problematic.

## CONCLUSIONS

This study builds on the established sociological literature with implications for public health messages. While the public have an unquestionable responsibility to try and use scarce emergency resources appropriately, merely informing them to 'only use emergency services in a genuine emergency' is unlikely to be of practical use in their moment of need. Where PCSCs enter ambulance workflows, there often exists a sequence of events where alternative avenues have been rationally explored but appear unsuitable. The public (and in particular, those calling on behalf of another) may need specific, detailed practical guidance to help them 'hold' some of the risk inherent in an uncertain situation. The present systems do not appear to permit the handing-back of control of the situation to caller. This may require a specific triage system that uses inherently different logic to 'first party' calls.

**Acknowledgements** The authors would like to acknowledge Nigel Rees and the Pre-Hospital Research Unit at the Welsh Ambulance Service NHS Trust for facilitating this study, and the staff and service users who participated.

**Contributors** MJB, SP and ARGS conceived and designed the study. MJB acquired and analysed the data. SP, ARGS and RB contributed significantly to the analysis and interpretation of the data. MJB drafted the manuscript. All authors revised the manuscript for important intellectual content and gave final approval for the version to be published.

**Funding** MJB was funded by a National Institute for Health Research (NIHR) Doctoral Research Fellowship. This paper presents independent research funded by the National Institute for Health Research (NIHR).

**Disclaimer** The views expressed are those of the authors, and not necessarily the NIHR, the NHS or the Department of Health.

**Competing interests** None declared.

**Patient consent for publication** Not required.

**Ethics approval** The study received a favourable opinion from South West (Frenchay) Research Ethics Committee (reference 15/SW/0307), and appropriate local governance approvals were obtained.

**Provenance and peer review** Not commissioned; externally peer reviewed.

**Data availability statement** No data are available.

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
