## [Reviewer comments · BMJ Open]

ARTICLE DETAILS

TITLE (PROVISIONAL)	Ambulance use for 'primary care' problems: an ethnographic study of seeking and providing help in a UK ambulance service
AUTHORS	Booker, Matthew; Purdy, Sarah; Barnes, Rebecca; Shaw, Ali

VERSION 1 – REVIEW

REVIEWER	Ricardo Angeles McMaster University, Canada
REVIEW RETURNED	01-Aug-2019

GENERAL COMMENTS	The paper is a good addition to the body of knowledge regarding why patients with primary care sensitive conditions call emergency services. Most of the paper is well written. However the abstract does not capture the essence of the results. The results were clear though some need a bit of explanation (I understood more as I read the paragraphs underneath each themes) to understand fully. Words like "watershed" are not necessary, arbitrary timeline or treatment expectations should be enough. But my concern is that the main results are not in the abstract.
---

REVIEWER	A N Siriwardena University of Lincoln, UK
REVIEW RETURNED	23-Aug-2019

GENERAL COMMENTS	Thank you for asking me to review this study. The authors conducted an ethnographic study of seeking and providing help in a single UK ambulance service through observations, interviews and documentary analysis. Overall the study is well conducted and presented and the limitations, particularly in relation to the selection of cases and use of a single ambulance service and investigator are discussed. I think this paper would be improved with a fuller discussion of how the 'new' concepts in the analysis relate to previous very similar concepts which have been already described in the sociological literature. For example, 'An arbitrary deadline or watershed is reached (internal factor)' is very similar to David Mechanic's concept of temporization. The idea of 'An outside offers advice' relates to the concept of 'the lay referral network' described by Friedson. In relation to this, the people referred to as 'outside' are often relatives or friends, so they may not be 'outsiders' from the perspective of the patient – this should be discussed. There are other parallels with the development of social theory related to 'illness behaviour' which the discussion would benefit from. Sometimes blocks 'An alternative avenue of care meets a block (external factor)' are presented by the patient or relative as a
---

	means of accessing help via another route, i.e. they may not exist or are exaggerated. Minor issues: There are a number of typos and grammatical errors in the text which should be corrected.
--	---

VERSION 1 – AUTHOR RESPONSE

Reviewer 1:

Most of the paper is well written. However the abstract does not capture the essence of the results. The results were clear though some need a bit of explanation (I understood more as I read the paragraphs underneath each themes) to understand fully. Words like "watershed" are not necessary, arbitrary timeline or treatment expectations should be enough. But my concern is that the main results are not in the abstract.

Response: Thank you for highlighting that you felt, having read the paper, the abstract didn't fully capture the essence of the results. We have re-structured the results section of the abstract to hopefully present more clearly – within the permitted abstract word count – a headline summary of the three cross-cutting concepts that emerged from the analysis. We have changed some of the words slightly to try to be a little more succinct with what these three 'messages' are. The unnecessary word 'watershed' has been removed.

Reviewer 2:

I think this paper would be improved with a fuller discussion of how the 'new' concepts in the analysis relate to previous very similar concepts which have been already described in the sociological literature. For example, 'An arbitrary deadline or watershed is reached (internal factor)' is very similar to David Mechanic's concept of temporization. The idea of 'An outside offers advice' relates to the concept of 'the lay referral network' described by Friedson. In relation to this, the people referred to as 'outside' are often relatives or friends, so they may not be 'outsiders' from the perspective of the patient – this should be discussed. There are other parallels with the development of social theory related to 'illness behaviour' which the discussion would benefit from. Sometimes blocks 'An alternative avenue of care meets a block (external factor)' are presented by the patient or relative as a means of accessing help via another route, i.e. they may not exist or are exaggerated.

Response: Thank you for these helpful suggestions with regards to the wider sociological / medical sociological literature. Such discussion was initially kept very brief due to manuscript length considerations, but taking into account the useful comments above about how this would enhance the discussion, we are pleased to refer to some broader sociological literature in the discussion. With the references above, we have written an additional paragraph in the discussion section, and reworded a couple of other paragraphs (and abstract/summary points) slightly to make reference to the ideas of Mechanic, Zola, Friedson alongside the theories of Anthony Giddens and Ulrich Beck. This hopefully articulates the point that these concepts – although well-recognised in the sociological literature - have direct application to the phenomenon of PCSCs in ambulance care. We hope this better supports the main thrust of the paper; strategies to mitigate rising demand for PCSCs need to be aware of the nuances that this data suggests.

There are a number of typos and grammatical errors in the text which should be corrected.

Thank you – the paper has been re-proofed with correction of typographical errors identified.